# Use of an In Vitro Skin Parallel Artificial Membrane Assay (Skin-PAMPA) as a Screening Tool to Compare Transdermal Permeability of Model Compound 4-Phenylethyl-Resorcinol Dissolved in Different Solvents

**DOI:** 10.3390/pharmaceutics13111758

**Published:** 2021-10-21

**Authors:** Bálint Sinkó, Vivien Bárdos, Dániel Vesztergombi, Szabina Kádár, Petra Malcsiner, Anne Moustie, Chantal Jouy, Krisztina Takács-Novák, Sebastien Grégoire

**Affiliations:** 1Department of Pharmaceutical Chemistry, Semmelweis University, H-1092 Budapest, Hungary; bsinko@pion-inc.com (B.S.); bardos.vivien@phd.semmelweis.hu (V.B.); vesztdani@gmail.com (D.V.); malcsiner.petra@gmail.com (P.M.); 2Pion Inc., Billerica, MA 01821, USA; 3Department of Organic Chemistry and Technology, Budapest University of Technology and Economics, H-1111 Budapest, Hungary; kadar.szabina@vbk.bme.hu; 4L’Oréal Research & Innovation, 93601 Aulnay-sous Bois, France; anne.moustie@rd.loreal.com (A.M.); chantal.jouy@rd.loreal.com (C.J.); sebastien.gregoire@rd.loreal.com (S.G.)

**Keywords:** Skin-PAMPA, formulation, skin barrier, permeability, pig skin, safety testing

## Abstract

Absorption through the skin of topically applied chemicals is relevant for both formulation development and safety assessment, especially in the early stages of development. However, the supply of human skin is limited, and the traditional in vitro methods are of low throughput. As an alternative, an artificial membrane-based Skin Parallel Artificial Membrane Permeability Assay (Skin-PAMPA) has been developed to mimic the permeability through the stratum corneum. In this study, this assay was used to measure the permeability of a model compound, 4-phenylethyl-resorcinol (PER), dissolved in 13 different solvents that are commonly used in cosmetic formulation development. The study was performed at concentrations close to the saturated solution of PER in each solvent to investigate the maximum thermodynamic potential of the solvents. The permeability of PER in selected solvents was also measured on ex vivo pig skin for comparison. Pig ear skin is an accepted alternative model of human skin. The permeability coefficient, which is independent of the concentration of the applied solution, showed a good correlation (*R^2^* = 0.844) between the Skin-PAMPA and the pig skin permeation data. Our results support the use of the Skin-PAMPA to screen the suitability of different solvents for non-polar compounds at an early stage of formulation development.

## 1. Introduction

Absorption through the skin of topically applied chemicals (e.g., drugs, cosmetics, iatrogenic substances) is relevant for both formulation development and safety assessment [1,2]. In the pharmacological domain, transdermal drug delivery offers multiple advantages over oral or parenteral administrations (e.g., by-passing “first-pass” metabolism, providing sustained drug release, protection of the GI tract from drugs, fewer side effects) [3]. In the cosmetic industry, the safety assessment of ingredients requires an estimation of their local and systemic exposure(s) when applied topically. Guidelines define clear criteria to conduct such skin absorption studies and point out in vitro human skin as the gold standard for study or pig skin as an alternative [4,5]. Although many data are available in these guidelines, the quality and reproducibility of the data are related to the assay criteria defined in the guidelines (skin preparation, receptor fluid chosen, skin test integrity, etc.) [6,7] and also to the validation of the analytical methods [8]. As the evaluation of skin penetration of compounds is needed at an early stage of development, such skin absorption study on ex vivo human skin is not suitable. As an alternative model, reconstructed skin has been utilized [9], with some limitations on the reproducibility and prediction capacity [10]. Alternatively, synthetic membrane models have been developed to mimic the main features of the stratum corneum (SC) [11,12,13,14,15], which acts as a rate-limiting barrier [16]. These membranes are easily available and are more cost-effective than ex vivo human skin. Moreover, it has been already demonstrated that such models can be successfully used in an initial screening approach to assist formulation selection before a more biological model is involved [17,18].

Recently, an artificial membrane-based in vitro method, the Skin Parallel Artificial Membrane Permeability Assay (Skin-PAMPA), was developed in a 96-well plate format [15]. Such layout is suitable for automation as well as high-throughput screening. This Skin-PAMPA model has been shown to possess a high prediction capability [19] not just for buffer based sample solutions, but also for both semisolid formulations (gel, ointment and cream) [20,21] and transdermal patches [22].

In product formulation, various vehicles are designed to modulate skin absorption by altering the solubility and permeability of an active ingredient. Penetration across the SC involves interactions among the solvent(s), SC and the active ingredient. Even if an artificial membrane cannot mimic SC in its overall complexity, it could be used to investigate the effect of solvent itself. Therefore, this project aimed to investigate the applicability of the Skin-PAMPA model on a wide range of safe “solvents” traditionally used by the cosmetic industry. In the study, the permeation of a model compound, 4-phenylethyl-resorcinol (PER) (see structure in Table 1), a skin-lightening agent used both in cosmetic and dermatologic formulations, was tested on the Skin-PAMPA model in 13 solvents (9 pure solvents and 4 simple mixtures, coded as S1–S13 in Table 2). The model compound was selected based on three aspects: (i) physico-chemical properties, (ii) solubility in a wide range of relevant solvents (to some degree) and (iii) good UV absorption to make the direct UV spectroscopy possible. PER is a non-polar, weak acid that is neutral at physiological pH. It has suitable UV properties and reasonable solubility in the solvents examined, which made it a good model for the study. A recently published study from Zhang and co-workers [23] has reported a comprehensive characterisation of PER, including HPLC-based *logP* and solubility and in vitro permeation studies through human and porcine skin. The permeation profile of PER was investigated in finite dose conditions using Franz diffusion cell method and applying PER in three different vehicles. The study concluded that the properties of PER make it a suitable compound for dermal delivery, which also confirms our selection of PER as model compound for this study.

The aqueous solubility of PER was measured and compared with available data. To have comparable results between the different solvents used, PER was solubilized at saturation in tested solvents. Infinite conditions were used for all experiments, as this allowed measuring typical parameters describing percutaneous absorption [24]: permeability coefficient (*P_m_* for PAMPA and *K_p_* for pig skin permeation), flux (*J*) and the amount penetrated in a finite time (*Qt*). To identify the best parameter to differentiate between the percutaneous absorption of PER in different solvents and to validate the Skin-PAMPA, the samples were also tested in a pig skin model. This method was suggested as a suitable alternative to human skin by the Scientific Committee on Consumer Safety [4]. Since pig skin penetration assays are resource- and time-consuming, it was not possible to measure the penetration of PER in all solvents. Hence, a limited number of solvents (9 out of 13) were tested spanning different types and solubility. In addition, not all formulations were suitable for this assay, since a sufficiently high concentration could not be achieved, due to the low solubility (S11, S12, S13).

**Table 2 pharmaceutics-13-01758-t002:** Solvents used in the study and the solubility of PER in different solvents.

Solvent Class	Code	Solvent	MW	PER Approximate Solubility ^1^ (mg/mL)	PER Equilibrium Solubility ^2^ (mg/mL)
Low-MW polar solvents	S1	Water	18.0	1	1.3 ± 0.2
	S2	Ethanol	46.1	>1000	368 ± 52
	S3	Glycerol	92.1	5	-
	S4	Dimethylisosorbide	174.2	75	60 ± 5.7
	S5	Water/ethanol 80:20 (*w*/*w*)	NA	10	8.1 ± 4.3
	S6	Water/dimethylisosorbide 90:10 (*w*/*w*)	NA	1	1.1 ± 0.1
Low-MW polar “glycol” solvents	S7	Propylene glycol	76.1	500	350 ± 21
	S8	Water:propylene glycol 80:20 (*w*/*w*)	NA	10	5.1 ± 0.8
	S9	Water/propylenglycol/ethanol 10:30:60 (*w*/*w/w*)	NA	>1000	373 ± 49
High-MW non-polar solvents	S10	Capric/caprylic triglycerides	554.8/470.7	75	74 ± 5.1
	S11	Octyl dodecanol	298.6	1	-
	S12	Apricot kernel oil	NA	1	-
	S13	Corn oil	NA	1	-

^1^ Approximate solubility of PER determined by semi-quantitative method [25], ^2^ equilibrium solubility (*S*_o_) measured with LC/MS/MS for the pig skin studies.

## 2. Materials and Methods

### 2.1. Materials

All solvents were provided by the French L’Oréal Laboratories. PER (CAS 94-77-9) was obtained from Symrise™ (Table 1). Lucinol (CAS 18979-61-8) used as internal standard for LC/MS-MS PER quantification was provided by L’Oreal. The applied concentrations of PER in different solvents are shown in Table 2. For sake of simplicity, the solvents are referred to in the text using a code system S1–S13. All the other reagents were of analytical grade and purchased from Sigma-Aldrich (Lyon, France) or Reanal™ (Budapest, Hungary). Pig ear skin was obtained from a slaughterhouse (Pouldreuzic, France), frozen at −20 °C after sampling and stored prior to use.

### 2.2. Solubility Measurements

As a first step, the solubility class at 32 °C (i.e., the temperature of the human skin surface) was determined according to the OECD test guideline No. 105 [25]. In a stepwise procedure, increasing volumes of the given solvent (pre-warmed at 32 °C) were added to precisely weighted amount 0.1 g of the PER sample in a 10 mL glass-stoppered measuring cylinder. After each addition of the solvent aliquots, the mixture was shaken for 10 min and evaluated visually for any undissolved particles of the solid. When, after addition of 10 mL of solvent, the sample remained undissolved, the experiment was continued in a 100 mL cylinder. The approximate solubility is given as the volume of the solvent in which complete dissolution was observed after 1 h. The sample was then stirred for 24 h before a final visual assessment. Based on this method, five solubility categories were set between 1 and 1000 mg/mL. A further refinement step included four subclasses in each category. For example, if the compound was soluble in the 1–10 mg/mL category, the solution was checked at concentrations of 2.5, 5, 7.5 and 10 mg/mL to determine the closest value to the saturated solution. These solubility categories defined the concentrations in the permeability test, which had a maximum of 500 mg/mL (50% of the maximum solubility category: 1000 mg/mL). For solvents tested on pig skin, solutions used were analysed by LC/MS/MS. For this purpose, a solution at an upper limit of the solubility class previously defined was prepared and centrifuged at 14,000 rpm to guarantee particle precipitation before analysis.

The equilibrium intrinsic solubility value of model compound in the acceptor medium was determined by the standardized protocol of saturation shake flask method [26,27]. The measurements were carried out at a controlled temperature 32.0 ± 0.5 °C. The sample was added to 5 mL of Prisma buffer solution pH 7.4 (which served as the acceptor phase in PAMPA experiments) until a heterogeneous system (solid sample and liquid) was obtained. The solubility suspension containing solid excess of the sample was stirred for a period of 6 h (stirring time) followed by 18 h of sedimentation to achieve the thermodynamic equilibrium. After sedimentation and the necessary dilution, the concentration of the saturated solution was measured by UV spectroscopy. The solubility experiments were performed in triplicate.

### 2.3. LogP Measurement

The *logP* value of PER was measured in octanol/water system at 25.0 ± 0.1 °C by standard shake-flask method described in our former papers [28,29]. Two parallel experiments were carried out.

Four different phase ratios of octanol:water (1:50, 1:75, 1:100, 1:125) were applied. The equilibration time was 1 h (Lauda M2OS, shaking thermostat Königshofen, Germany), and the phases were separated by centrifugation. The concentration decrease in the sample in the aqueous phase was detected by UV spectroscopy (Jasco V-550 UV/VIS spectrophotometer, Easton, MD, USA) measuring the absorbance before and after the partition at $\lambda_{\max}$= 280 nm. The *logP* value was calculated from the equation:(1)$$logP=log\left[\frac{A_{0}-A_{1}}{A_{1}}\left(\frac{V_{aq}}{V_{oct}}\right)\right]$$
where *A*_0_ and *A*_1_ represent the absorbance value at the absorption maximum of the compound in the aqueous phase before and after partition [28].

### 2.4. Permeability Measurements Using Skin-PAMPA Plates

Membrane permeability of PER was measured using commercially available Skin-PAMPA plates (Skin-PAMPA™, Pion Inc., Billerica, MA, USA). Skin PAMPA™ sandwiches and stirring bars (P/N: 110211) were supplied by Pion Inc™. UV plates (UV-star microplate, clear, flat bottom, half area) were from Greiner Bio-one™ (Kremsmünster, Austria). Membranes were hydrated overnight with standard hydration solution (Pion Inc™., product number 120706). The donor phase solutions of PER in different solvents were prepared freshly according to the approximate solubility (Table 2), and 70 μL (corresponding to 233 µL/cm² for 0.3 cm² exposure area) was applied to the donor (upper) plate. The acceptor (lower) plate contained 180 μL Prisma buffer pH 7.4 and a magnetic stirrer in each well. The PAMPA™ sandwich was incubated at 32 °C in a Gut-Box™ (from Pion Inc™). Stirring bars were applied in every well to avoid the effect of the unstirred water layer. The acceptor solution was sampled after 7.5, 15, 30, 60, 120, 240 and 360 min incubation. After each individual incubation period, 150 µL from the acceptor compartment was transferred to UV plates. The acceptor phase was replaced with fresh buffer solution. UV absorption was measured at λ = 280 nm (Tecan Infinite M200 UV-plate reader driven by Magellan v.7.2. software (Tecan™, Männedorf, Switzerland) after dilution if necessary, and the concentration of PER was calculated using the calibration curve A = 117.95 *c* + 0.01 (*R*^2^ = 0.9997, *n* = 9), in the concentration range 9–90 µg/mL.

Parameters characterizing the transdermal penetration were obtained from the cumulative amount of PER penetrated per cm^2^ versus time plots. The flux (*J)* was obtained as the slope of the permeability profile and expressed in µg/cm^2^ × h units. For the linear regression analysis, the linear range of incubation period from 0 to 30 min was selected and used for calculation of flux of the model compound. Permeability coefficient *P_m_* (cm^−2^ × h^−1^) was calculated from the equation:*P*_*m*_= *J*/*C*_*D*_(2)
where *C_D_* is the donor phase concentration.

The area under the curve (*AUC*) was calculated by integration of the permeability profile between 0 and 6 h using OriginPro v.2019b (OriginLab Corporation, Northampton, MA, USA).

### 2.5. Skin-PAMPA Membrane Integrity Study

Possible disruption by solvents of the integrity of the biomimetic artificial membrane was investigated. Wells were filled with each solvent and incubated over a longer (minimum 7 h) incubation time than the duration of the tests with the model solutions. The solvents were aspirated from the wells, and the residue from the surface of the membrane was removed gently with cotton paper. A standard skin permeability assay was then performed using piroxicam as the model permeant, for which precise previous data are available [22]. The *logP_m_* values were compared with the reference value from untreated plates.

### 2.6. Penetration Kinetics across Pig Ear Skin

Before use, hairs were shaved from the pig ear skin using an electric razor, and the skin thickness was adjusted between 700 and 1200 µm. This size range was achieved by cutting the dermis below hair follicle. The integrity of the skin was tested according to the Trans-Epidermal Water Loss (TEWL) method using a Delfin device. The TEWL of dermatomed skin was always lower than 15 g/m^2^ × h (cut-off value was defined according to historical date obtained in the lab), indicating that storage at −20 °C and dermatome did not compromise skin integrity. The number of discs per treatment was between 2 and 10 replicates.

After topical application of the test chemical (infinite dose, 1.13 mL/cm^2^), the concentration of the chemical in the receptor fluid was measured by sampling 200 µL of receptor fluid and replacing it with fresh fluid on an hourly basis, up to 16 h. The receptor fluid selected for PER was sodium chloride solution (9 g/L) supplemented with 0.25% (*v*/*v*) Tween80.

The kinetic samples were directly injected into an LC/MS-MS system (Shimadzu Nexera LC system, Shimadzu, Kyoto, Japan) coupled with a mass spectrometer API 3500 (Sciex, Framingham, MA, USA). The analytical system was managed by Analyst v.1.6 software (Sciex, Framingham, MA, USA). The analytical column used was a Kinetex C18 from Phenomenex™ (Torrance, CA, USA) (50 × 2.0 mm, dp. 2.6 µm), and analysis was carried out with a gradient elution with mobile phases of 20 mM ammonium acetate (A) and acetonitrile (B). The column temperature was fixed at 50 °C, and the volume of the injection was 10 µL with a flow rate of 0.8 mL/min. The ionisation mode used was electrospray negative. MRM was used for detection with the transitions 213 → 198.2 for PER and 165→121 for Lucinol as internal standard.

The specificity of the analytical method was controlled with blank (NaCl, 9 g/L) solution (Merck, Darmstadt Germany). The limit of quantitation (LoQ) was 2.43 ng/mL. Linearity was determined between the LoQ and 1000 ng/mL, with accuracy below ±15%, except at the LoQ, which was below ± 20%. Accuracy and precision were determined at least at two quality control (QC) theoretical concentrations: low (around 20 ng/mL) and middle (around 300 ng/mL). All QCs remained within the acceptance criteria (accuracy < ±15%). Matrix effects and stability in buffer solutions and buffer supplemented with pig skin were evaluated at two concentrations (426 and 21.5 ng/mL) in triplicate by spiking buffer solutions containing known amounts of chemical. The stability in buffer solutions spiked with PER was 98.3 ± 7.0%. A matrix effect was observed; therefore, all calibrations for this chemical were carried out in the matrix.

The penetration parameters (permeability coefficient, *K_p_*, and flux) were determined from the curves representing the cumulative amount per unit area of skin (*Qt*, μg/cm^2^) as a function of time (h). The calculation was carried out using GraphPad PrismT v.7 (GraphPad Software Inc., San Diego, CA, USA).

## 3. Results

### 3.1. Solubility of PER in Different Solvents

Approximate solubility at saturation was measured for 13 solvents with the method described in the Methods section. In addition, the solubility was measured independently with LC/MS/MS methods for nine solvents to confirm the validity of the semi-quantitative approach. For these solvents, the differences in solubility values between the two different approaches were within a factor of 2, except for the highest solubility (i.e., PER in S2 and S9). This means good agreement, as shown by Table 2. Zhang and co-workers [23] have recently reported PER solubility data in propylene glycol (PG), glycerol and dimethylisosorbide (DMI). The reported values for PG are in good agreement with the results found in this study, but the data for DMI and glycerol are significantly different, which may be explained by the differences in their method, in the amount of solid excess or in the crystal form.

The solvents are grouped into three main types according to their molecular weight (MW) and polarity (see Table 2). The first group included six low-MW polar solvents; the second group included three low-MW polar “glycol” solvents; and the third group included four higher-MW non-polar solvents. Four solvents were simple two- or three-component solvent mixtures. The approximate solubility classification of PER at 32 °C was in agreement with its lipophilicity. PER, with a *logP* of 2.98, is poorly soluble (~1 mg/mL) in water and in highly non-polar organic solvents (S11–13), while it is readily soluble in semi-polar organic solvents (S2, S7, S9). 

The solubility at 32 °C in Prisma buffer pH 7.4 was also measured. PER is present at this pH in non-ionized form; thus, the value obtained is the intrinsic solubility (*S_o_*). The intrinsic solubility of PER was found to be 3.45 ± 0.01 mg/mL. Ten percent of this value, 0.345 mg/mL, has been selected as the target upper limit of the concentrations in the acceptor compartment to maintain a steady-state sink condition throughout the assay.

### 3.2. Effect of Solvents on PAMPA Membrane Integrity

No solvent effect was recorded on the visual appearance of the membranes after their removal and before the addition of piroxicam solution. The *logP_m_* of piroxicam from aqueous solution measured across each solvent-treated membrane ranged between −3.82 and −4.81, with a mean of −4.25 ± 0.30 (Figure 1). This fits well with the reference *logP_m_* value of −4.98 ± 0.01 that was measured previously in an aqueous solution using this PAMPA model [22]. All permeability values of piroxicam were within one order of magnitude of the previous control *logP_m_* value. Variation, i.e., the SD of the permeability values of piroxicam, provides a good indication of membrane integrity. Extreme high standard deviation would indicate membrane damage. As shown in Figure 1, the error bars are small (average SD ± 0.08), with ethanol (S2) presenting the highest variation (SD: ± 0.23) and thus the largest effect on the membrane, but this SD is still acceptable, indicating an interaction of ethanol with the membrane rather than its corruption.

Therefore, all solvents were considered appropriate for the study, with minor signs of membrane interaction.

### 3.3. Effect of Solvent on the Permeability of PER Using PAMPA

PAMPA measurements aim to provide relevant information regarding the effect of solvents on skin permeability in a high-throughput screening format; therefore, the most characteristic parameters of the permeation process (see Table 3) were calculated from the permeated amount vs. time plot. In the case of infinite dosing conditions, the permeated amount vs. time plot is expected to be linear up until the point where the acceptor concentration is reaching the limitation of solubility or inifinite dose is no longer respected (i.e., concentration in donor compartment significantly decreased) on the example profile of PER dissolved in water on Figure 2a (solvent S1). Similar saturation curves and linear regression plots were obtained for all the solvents (see Appendix A). To avoid the impact of the limitations, the first three timepoints (7.5 min, 15 min, 30 min) were selected to calculate the flux through the membrane and the lag time. Figure 2b shows the result of the linear regression analyses. The linear range of the permeability profile was used for the calculation of flux (slope of the linear equation) and lag time (*x* value at *y* = 0). The calculated lag times (0–6 min) indicated fast membrane saturation of PER regardless of the solvent.

The relevant permeability parameters across Skin-PAMPA membranes of the non-polar test chemical, PER, dissolved in different solvents are shown in Table 3. The different solvents had a significant impact on its permeability, whereby the *logP_m_* values varied by about 1.5 orders of magnitude (which is more than the variation caused by the solvent alone). The *highest* permeability (*logP_m_* > −1.2) was achieved when PER was dissolved in water (S1) and predominantly water-containing mixtures (S5, S6 and S8). Solvents resulting in *medium* permeability (*logP_m_* range −1.5 and −2.3) mainly covered two chemical types: (a) higher-MW non-polar solvents such as long-chain fatty acid esters (S10) and long-chain alcohol (S11), and (b) the small polar alcohols (S2, S3) and a solvent mixture containing glycol and ethanol (S9). Solvents resulting in *low* permeability (*logP_m_* < −2.4) of PER were low-MW polar organic solvents including dimethylisosorbide (S4) and propylene glycol (S7), in which PER was readily soluble (75 and 500 mg/mL, respectively). There were two higher-MW non-polar solvents in which PER was poorly soluble (~1 mg/mL), namely, apricot kernel oil (S12) and corn oil (S13).

### 3.4. Comparison of the Permeability of PER Using Skin-PAMPA vs. Pig Skin

The permeability of PER across pig skin was determined for nine solvents (see data in Table 4) and compared with the values measured in the PAMPA. These included mostly low-MW polar solvents (S1, S2, S4, S5, S6), three low-MW polar “glycols” (S7, S8, S9) and one high-MW non-polar solvent (S10).

Figure 3a shows the comparison of flux between Skin-PAMPA and pig skin (in increasing values for pig skin). The absolute values of flux in Skin-PAMPAs were higher than for pig skin, but a comparison can be done by showing the values on different y-axes. In four solvents (S1, S5, S6 and S8), the difference was within one order of magnitude despite potential underestimation of permeability coefficient using Skin-PAMPA. In contrast, for three solvents (S4, S7 and S10), Skin-PAMPA overestimated PER flux by more than two orders of magnitude. There were two clear outliers in the correlation: solvents S2 and S9.

Permeated amount at 6 h, expressed as *AUC_PAMPA_* (calculated by integration of the permeability profile between 0 and 6 h and normalized to the donor concentration) for Skin-PAMPA and *Qt_pig skin_* at 16 h for pig skin, was also correlated. The comparison between the amounts of PER detected in the acceptor compartments in PAMPA and pig skin provides a much closer trend than flux data comparison (Figure 3b). Only two outliers were detected, namely S1 and S6 (their data not shown), and for seven solvents, the correlation coefficient is *R*^2^ = 0.843, which can be considered reasonable.

The best correlation between the two models occurred when the data were expressed as the *logP_m_* for PAMPA and *logK_p_* for pig skin assays (Figure 3c). Since these values were normalised to the concentration in the donor compartment, they can be plotted on the same y-axis. The calculation of the two permeability values applied the same mathematical equation derived from the relationship *P_m_* = *J*/*C_D_*, where *J* is the flux, and *C_D_* is the initial donor concentration. As expected, when comparing a single membrane with multiple layered 700–1200 µm thick native pig skin, the permeability coefficients in Skin-PAMPAs were higher than pig skin. The biggest differences between values from the Skin-PAMPA and pig skin models were when PER was dissolved in dimethylisosorbide (S4) and ethanol (S2). Nevertheless, there was a good correlation between the two values for the nine solvents, with an *R*^2^ of 0.844. In fact, there was no outlier in this correlation between the two models.

## 4. Discussion

The applicability of the Skin-PAMPA was investigated as a screening tool to differentiate between the permeabilities of a model compound dissolved in different solvents. The study was focused on the behavior of the Skin-PAMPA membrane when an active ingredient was applied in different solvents that are applied routinely in the cosmetic industry. PER was selected as the model compound of non-polar chemicals, which is a well-studied compound with a range of physico-chemical properties already available in the literature.

The Skin-PAMPA measures the permeability of solutions that are close to their saturated concentrations. Therefore, we measured the solubility of PER in each solvent. Since an exact solubility was difficult to measure for some solvents, solubility was classified in five main categories and four sub-categories based on visual assessments. The classification of solubility based on visual evaluation (of any undissolved particles of the solid) correlated very well with that measured using LC/MS/MS methods for the pig skin assays (Table 2). PER was soluble in the solvents tested, and the data were in agreement with its moderate lipophilicity, such that it was poorly soluble in highly non-polar and highly polar solvents, while it was best dissolved in semi-polar solvents and their mixtures.

When conducting skin penetration assays, technical aspects that could impact results should be considered. One of the important aspects to consider is that the permeated amount should not exceed 10% of the applied dose to provide accurate permeability values. When it exceeds 10% of the applied dose, the permeability coefficient may be underestimated. A second aspect relates to the effect of the solvent itself upon the integrity of the membrane. In order to investigate the direct effect of solvents on the Skin-PAMPA membrane integrity, the membrane was pre-treated with each solvent (in the absence of chemical), and after removing them, piroxicam was measured as the test permeant [22]. The permeability of piroxicam was increased depending on pre-treatment with the solvents, possibly due to the partitioning of piroxicam (*logP* = 1.71) into the residual solvent layer at the surface of the membrane, which provided a higher surface concentration. However, the variation in permeability of piroxicam in the different solvents was found to be of minor amplitude, which indicates that the membrane integrity was intact, so the solvents were not damaging the membrane structure, or at least all changes were stable by the end of the incubation.

The Skin-PAMPAs were performed with a 6 h incubation in each studied solvent solution, which allowed high-throughput evaluation. The permeability of PER was significantly affected by the solvent in which it was dissolved, such that the *logP_m_* spanned about 1.5 orders of magnitude. The permeability could be divided into three classes: *low* (*logP_m_* < −2.4), *medium* (*logP_m_* from −2.3 to −1.5) and *high* (*logP_m_* > 1.2). Solutions of PER provided examples for all classes.

Great attention had to be devoted to the following factors, which are the limitations of this method. Appropriate precise pipetting is essential in this technique. Compounds with excessively high or low permeation properties cannot be measured. Applying viscous solvents can be challenging because the application of solvents to PAMPA plate is a time-consuming process, so correction for the time factor needs to be implemented during the evaluation of the results. Finally, the tension of the solvents can also be a limiting factor, since the concentration of high-tension solutions can be modified during the experiment, leading to invalid permeability results.

To determine whether the Skin-PAMPA model provides an accurate estimation of permeability, the results were compared with those obtained from penetration studies using pig skin. Permeation potential can be expressed in a number of ways: the amount penetrated in a finite time (*Qt*); flux (*J*), representing the mean mass transfer through the membrane; and the permeability coefficient (*P_m_* or *K_p_*), reflecting the rate of penetration through the membrane. Therefore, the comparisons between Skin-PAMPA and pig skin permeability were also used to identify the best parameter to differentiate the permeability of the chemical in different solvents. When the ranking of the permeation potential of PER in different solvents was expressed as the flux, there was a poor correlation between values from the Skin-PAMPA and pig skin assays. Better correlation was found between the amounts penetrated (*AUC_PAMPA_* vs. *Qt_pig skin_*), but two solvents were outliers. The best correlation was achieved when permeability was expressed as log of the permeability coefficient, *logK_p_* or *logP_m_*. Both assays indicated that the permeability of PER in solvents S2, S4, S7, S9 and S10 was higher than when it was dissolved in the other four solvents. The biggest differences between values from the Skin-PAMPA and pig skin models were observed when PER was dissolved in dimethylisosorbide (S4) and in ethanol (S2). These differences were not due to the solvent per se, since this was excluded in the pre-tests; however, the combination of PER and solvent may have disrupted the PAMPA membrane structure, resulting in a higher permeability. The difference in the permeability coefficients between the two models was much less when these solvents were in mixtures with water containing a lower concentration of the organic component (e.g., S6 and S5).

Notably, the ranking of the permeation potential was different based on the expression of the data. For example, S1 and S6 received low rankings when data were expressed as the amount penetrated or the flux, but they were ranked among the highest ones when the *logK_p_* or *logP_m_* were used. Since flux is the product of permeability and the donor concentration, and the concentrations tested were near to saturated values, a higher solubility in the donor compartment may be expected to result in a proportional increase in the flux. This was generally reflected in the Skin-PAMPA flux values for PER (Figure 4a,b), albeit with some exceptions (e.g., the solubility of PER in water and corn oil were both about 1 mg/mL, but the flux was 17-fold lower in corn oil than when dissolved in water) that indicate the importance of complex solubility/dissolution and permeation studies. By contrast, the permeability coefficient, *logP_m_* or *logK_p_*, is independent of the concentration used, making it a more appropriate measure of permeability for chemicals that have large differences in solubility in different solvents.

## 5. Conclusions

In conclusion, the Skin-PAMPA allows the evaluation of the permeability of model compound dissolved in multiple and widely varying solvent types, from highly polar to highly non-polar, as well as mixtures of solvents. It was possible to classify the permeability of PER into 3 categories: low, medium and high. The most appropriate parameter for the comparison of permeability was the permeability coefficient, *logP_m_* or *logK_p_*, which is independent of the concentration of the solution applied. This is particularly important for chemicals that have large differences in solubility. The comparison of the relative permeability of PER in different solvents was confirmed by comparing the permeability coefficients with those measured in pig skin permeability assays. Our results support the use of the Skin-PAMPA for screening the suitability of different solvents for non-polar test compounds at early stages of product development.

## Figures and Tables

**Figure 1 pharmaceutics-13-01758-f001:**
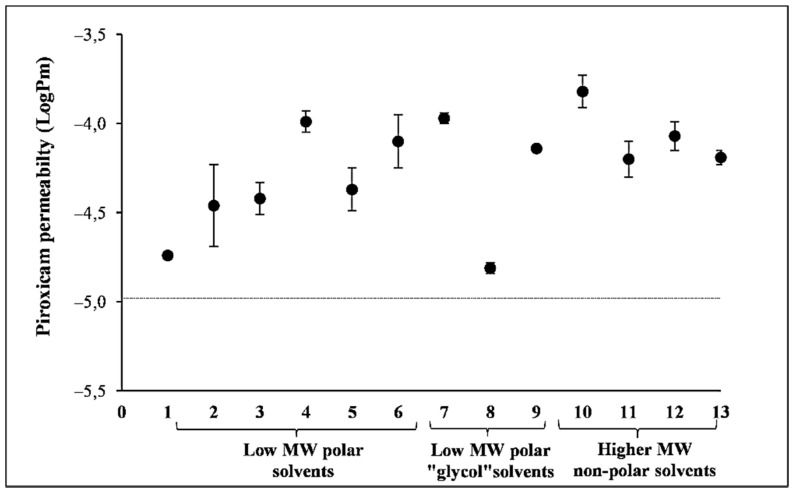
Effect of 13 solvents on the integrity of the Skin-PAMPA membrane using piroxicam as the model permeant. Permeability of piroxicam dissolved in water was measured after 7 h pre-treating membranes with each solvent. The permeability values are mean ± SD, *n* = 9.

**Figure 2 pharmaceutics-13-01758-f002:**
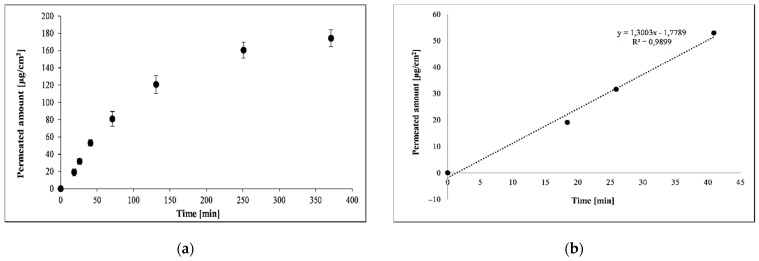
(**a**) The permeability profile of PER dissolved in water using Skin-PAMPA; (**b**) The linear regression curve used for the calculation of flux.

**Figure 3 pharmaceutics-13-01758-f003:**
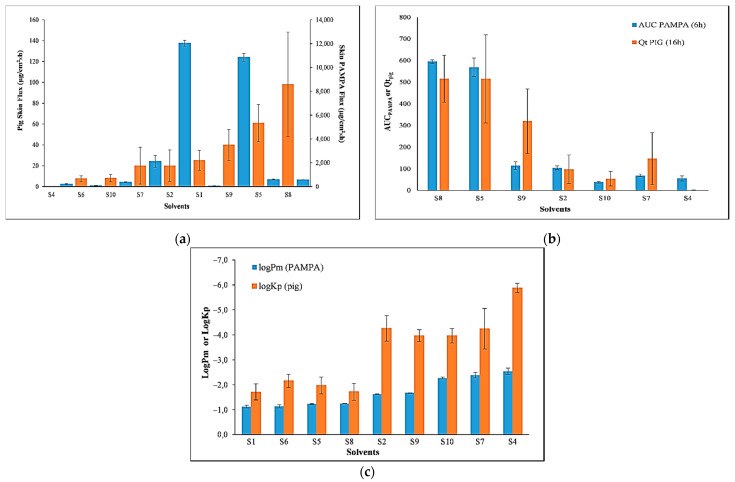
(**a**) Comparison of the permeability of PER in different solvents across Skin-PAMPA membranes (blue bars) and pig skin (orange bars), expressed as flux (no close correlation); (**b**) permeated amount (*R*^2^ = 0.834, *n* = 7); (**c**) permeability coefficient, *logP_m_* and *logK_p_* for Skin PAMPA and pig skin, respectively (*R*^2^ = 0.844, *n* = 9). All values are mean ± SD.

**Figure 4 pharmaceutics-13-01758-f004:**
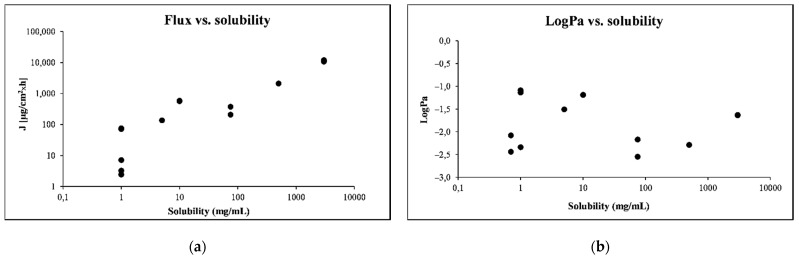
(**a**) Comparison of the solubility of PER with flux; (**b**) *logP_m_*. All values are mean.

**Table 1 pharmaceutics-13-01758-t001:** Chemical structure and physico-chemical properties of PER.

PER	
Structure	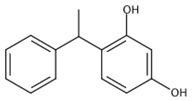
Chemical Name	4-phenylethyl-resorcinol
CAS Number	94-77-9
Molecular Weight	214.3 (g/mol)
*logP*	2.98 ^1^
*pK_a_*	9.77–10.77 (AH/A^−^) ^2^
Solubilty in Prisma Buffer	3.45 ^1^ (mg/mL)
Water Solubility	3.85 (mg/mL)

^1^ data measured at Semmelweis University; ^2^ data obtained from L’Oréal Laboratories.

**Table 3 pharmaceutics-13-01758-t003:** Characteristic parameters of permeability of PER across Skin-PAMPA membrane for comparison when applied in different solvents.

Code	*C_D_* [mg/mL]	*J* [µg/cm^2^ × h]	Lag Time [min]	Permeated Amount (6 h) [µg/cm^2^]	*AUC* Normalized to *C_D_*	*logP_m_*
S1	1	72.4 ± 7.8	1.4	169 ± 4	647	−1.12 ± 0.06
S2	500	12,033 ± 252	0.0	13,575 ± 1710	105	−1.62 ± 0.01
S3	5	137 ± 22	5.5	869 ± 112	491	−1.57 ± 0.07
S4	70	209 ± 53	3.6	1342 ± 298	55.2	−2.54 ± 0.13
S5	10	589 ± 25	0.9	1662 ± 169	569	−1.23 ± 0.03
S6	1	76.4 ± 15.2	0.0	175 ± 22	551	−1.13 ± 0.07
S7	500	2118 ± 502	3.8	11,142 ± 729	69.1	−2.38 ± 0.11
S8	10	570 ± 5	1.1	1772 ± 98	595	−1.24 ± 0.01
S9	500	10,846 ± 326	1.4	14,926 ± 2431	114	−1.66 ± 0.01
S10	70	377 ± 36	3.0	749 ± 92	38.2	−2.27 ± 0.04
S11	1	7.18 ± 1.6	3.9	19 ± 0.4	64.5	−2.16 ± 0.08
S12	1	2.45 ± 0.57	3.3	13 ± 1.9	36.7	−2.63 ± 0.07
S13	1	3.26 ± 0.41	1.4	13 ± 1.3	38.6	−2.48 ± 0.06

**Table 4 pharmaceutics-13-01758-t004:** Characteristic parameters of permeability of PER across pig ear skin in different solvents.

Code	C_D_ [mg/mL]	J [µg/cm^2^ × h]	Permeated Amount (16 h) [µg/cm^2^]	*logK_p_*
S1	1	25 ± 9.6	261 ± 100	−1.71 ± 0.32
S2	368	20 ± 15	98 ± 67	−4.26 ± 0.51
S4	60	0.08 ± 0.024	1.09 ± 0,39	−5.88 ± 0.19
S5	8	61 ± 18	515 ± 203	−1.97 ± 0.34
S6	1	7.5 ± 3	100 ± 37	−2.16 ± 0.26
S7	350	20 ± 17.6	147 ± 120	−4.24 ± 0.81
S8	51	98 ± 50	516 ± 109	−1.72 ± 0.34
S9	373	40 ± 15	320 ± 149	−3.97 ± 0.24
S10	75	8 ± 3.5	54 ± 34	−3.97 ± 0.29

## Data Availability

All data can be provided by the authors upon request. No publicly accessable archive storage is available.

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
