# Peer review of "Use of an In Vitro Skin Parallel Artificial Membrane Assay (Skin-PAMPA) as a Screening Tool to Compare Transdermal Permeability of Model Compound 4-Phenylethyl-Resorcinol Dissolved in Different Solvents"

_pharmaceutics, 2021, doi:10.3390/pharmaceutics13111758_

Round 1

Reviewer 1 Report

  • Table 2. it is better to be split approximate solubility & equilibrium solubility into 2 columns
  • The quality of all figures should be improved
  • Figure 2 repeated, which the 2nd one should be Figure 4 instead
  • In the title, They mentioned chemicals, while they used only one (PER)
  • Is it enough to use only one chemical to prove the theory?
  • why the authors selectively show the results of some solvents while not for others (Figure 3)?

Author Response

1. Table 2. it is better to be split approximate solubility & equilibrium solubility into 2 columns.

We thank the comment. The solubility data have been splitted into two columns for better clarity.

2. The quality of all figures should be improved.

Thank you for the suggestion. The graphs in better quality were exported and all figures were modified in the new version of manuscript.

3. Figure 2 repeated, which the 2nd one should be Figure 4 instead.

We apologize for the misprint. The correct numbering of figures was included in the new version of the manuscript.

4. In the title, They mentioned chemicals, while they used only one (PER).

We agree with this remark, and we changed the title. Chemicals” was modified to 4-phenylethyl-resorcinol. The new title of article:

Use of the in vitro Skin Parallel Artificial Membrane Assay (Skin-PAMPA) as a screening tool to compare transdermal permeability of model compound 4-phenylethyl-resorcinol dissolved in different solvents”

5. Is it enough to use only one chemical to prove the theory?

Thank you for this exiting question. We agree that to prove the applicability of Skin-PAMPA for prediction of human transdermal permeability requires investigation of several compounds with different chemical properties. Actually, it was done in our previous papers and has been confirmed by several studies by other authors. Therefore, the aim of this work was to investigate whether Skin-PAMPA is suitable to study the effect of solvents commonly used in dermatology and cosmetics on the transdermal permeability of non-polar molecules. The selection of the model compound (PER) was made with special attention since the physicochemical properties of PER represents well the topically applied non-polar drugs and cosmetics.

We have to note that a polar model molecule was also examined during measurements, but the results could not be reported in this paper due to the limit of spread.

6. Why the authors selectively show the results of some solvents while not for others (Figure 3)?

On Figure 3 only those 9 solvents are shown where permeability was studied by both methods. Since pig skin penetration assays are resource and time-consuming, it was not possible to measure the penetration of PER in all solvents. Hence, a limited number of solvents (9 out of 13) were tested in order to span different types and solubility. In addition, not all formulations were suitable for this assay since a sufficiently high concentration could not be achieved, due to the low solubility (S11, S12, S13).

Reviewer 2 Report

This manuscript studies the skin permeability of 4-phenylethyl-resor- 18 cinol dissolved in 13 types of solvents, and compare it with permeation from artificial membrane, which is commercially available. The pig skin was used for comparison. The study prosed that the membrane can be used for screening of solvents for non-polar molecules.

  • In general, manuscript lacks clarity in method and results. Need check for figure numbers, and more details in captions.
  • Abstract: initially it discusses the issue of human skin availability, where artificial membrane may be used as alternative. However, it is essential to compare with human skin instead of animal skin for true comparison before it can be used as alternative to human skin. Therefore, please revise the abstract.
  • Please clarify the statement “The study was performed close to saturation to investigate…”. What does saturation mean here?
  • Only lipophilic compound with ~200 g/mol molecular weight is used in the study, which cannot present other molecules with different log p and molecular weight. Ideally, the study comparing artificial membrane must include several molecules with different physicochemical properties for permeation. this study must include data comparing at least one more molecule which is far different in physiochemical properties of 4-phenylethyl-resor- 18 cinol. To specifically claim for non-polar molecules, the study must validate the results with other molecules.
  • The permeation profiles for membrane and skin are not presented. Please must provide all the permeation profiles. It may be included as supplementary.
  • Figure 2: please provide further details caption. Is this profile for pig skin or artificial skin?
  • Figure 1: please specify number or n value.
  • Figure 1: How this figure presents the integrity of membrane? Also include the data for integrity of pig skin.
  • What was the thickness of pig skin? Also, provide the anatomical location of pig skin.
  • Page 12, Figure 2a
  • Table 3, it includes data for artificial membrane. Please also provide data for pig skin.
  • Figure 3: Please include statistical comparisons.
  • How was the AUC calculated?
  • Conclusion: “…..chemicals dissolved in multiple and widely varying solvent types” please correct as only single chemical was investigated.
  • Please include the limitation of this study in the discussion section.

Author Response

This manuscript studies the skin permeability of 4-phenylethyl-resorcinol dissolved in 13 types of solvents, and compare it with permeation from artificial membrane, which is commercially available. The pig skin was used for comparison. The study prosed that the membrane can be used for screening of solvents for non-polar molecules.

1. In general, manuscript lacks clarity in method and results. Need check for figure numbers, and more details in captions.

Thank you for the helpful remark, the correct numbering of figures was included in the new version of the manuscript and the method and results sections are corrected for better clarity according to the comments below.

2. Abstract: initially it discusses the issue of human skin availability, where artificial membrane may be used as alternative. However, it is essential to compare with human skin instead of animal skin for true comparison before it can be used as alternative to human skin. Therefore, please revise the abstract.

Thank you for the remark. In our previous papers Skin-PAMPA results were compared with human skin permeability data to prove the applicability of the artificial membrane method. Since human skin supply is limited in this study we compared the data with pig ear skin model which is a suggested alternative of human skin. The abstract has been corrected.

“The permeability of PER in selected solvents were also measured on ex vivo pig skin for comparison. Pig ear skin is an accepted alternative model of human skin. The permeability coefficient which is independent from the concentration of the applied solution showed good correlation (R2=0.844) between Skin-PAMPA and the pig skin permeation data.

3. Please clarify the statement “The study was performed close to saturation to investigate…”. What does saturation mean here?

The questionable text was clarified in the abstract.

The study was performed at concentrations close to the saturated solution of PER in each solvent to investigate the maximum thermodynamic potential of the solvents.”

4. Only lipophilic compound with ~200 g/mol molecular weight is used in the study, which cannot present other molecules with different log p and molecular weight. Ideally, the study comparing artificial membrane must include several molecules with different physicochemical properties for permeation. this study must include data comparing at least one more molecule which is far different in physiochemical properties of 4-phenylethyl-resorcinol. To specifically claim for non-polar molecules, the study must validate the results with other molecules.

Our answer to this exiting question is the same as given for Reviewer #1 at point 5.

We agree that to prove the applicability of Skin-PAMPA for prediction of human transdermal permeability requires investigation of compounds with different chemical properties. Actually, it was done in our previous papers and has been confirmed by several studies by other authors. Therefore, the aim of this work was to investigate whether Skin-PAMPA is suitable to study the effect of solvents commonly used in dermatology and cosmetics on the transdermal permeability of non-polar molecules.  The selection of the model compound (PER) was made with special attention since the physicochemical properties of PER represents well the topically applied non-polar drugs and cosmetics.

We have to note that a polar model molecule was also examined during measurements, but the results could not be reported in this paper due to the limit of spread.

5. The permeation profiles for membrane and skin are not presented. Please must provide all the permeation profiles. It may be included as supplementary.

Thank you for the remark. On Figure 2 permeation profile of PER dissolved in water is shown representatively, all the other permeability profiles obtained in different solvents have been inserted into supplementary material.

6. Figure 2: please provide further details caption. Is this profile for pig skin or artificial skin?

The caption has been completed.

“Figure 2. (a) The permeability profile of PER dissolved in water using Skin-PAMPA”

7. Figure 1: please specify number or n value.

Thank you for the comment. The figure legend was completed with this information.

“Figure 1. Effect of 13 solvents on the membrane integrity of the Skin-PAMPA membrane using piroxicam as model permeant. Permeability of piroxicam dissolved in water was measured after 7h pre-treated membranes with each solvent. The permeability values are mean ± SD, n=9.”

8. Figure 1: How this figure presents the integrity of membrane? Also include the data for integrity of pig skin.

We agree that this topic needs more explanation.

The common membrane integrity methods are used to investigate the membrane stability upon storage. In this study our aimed was to reveal the possible damaging effect of organic solvents on the structure of the membrane.  No literature method is available for this purpose therefore, as described in the experimental section, first we visually assessed the membrane condition after the pre-treatment with different solvents. Afterward, the permeability of piroxicam from aqueous solution (a compound used previously in our laboratory for Quality Control of Skin-PAMPA) was measured on pre-treated membrane. We consider that extreme high permeability value would indicate the damage of the membrane. Furthermore, the low standard deviation of the results also shows the good membrane integrity. The text was extended for clarity.

Extreme high standard deviation would indicate the membrane damage. As shown in Fig. 1, the error bars are small (average SD ± 0.08), ethanol (S2) presenting the highest variation (SD: ± 0.23) and thus the largest effect on the membrane, but this SD is still acceptable indicating rather an interaction of ethanol with the membrane than its corruption.

For pig skin, as reported paragraph 2.6, pig skin integrity is controlled with Trans-Epidermal Water Loss (TEWL).

9. What was the thickness of pig skin? Also, provide the anatomical location of pig skin.

The anatomical location of the skin was the ears of the pig. The thickness of pig skin was between 700 and 1200 µm in the measurements.

10. Page 12, Figure 2a

We apologize for the misprint. The figure numbering was corrected.

11. Table 3, it includes data for artificial membrane. Please also provide data for pig skin.

Thank you for this suggestion. The manuscript was completed with Table 4, which contains the results of the pig skin measurement.

12. Figure 3: Please include statistical comparisons.

No close correlation was observed between the flux values  . In the case of the other two parameters the result of statistical comparison has been inserted to the figure legends.

Figure 3. (a) Comparison of the permeability of PER in different solvents across Skin-PAMPA membranes (blue bars) and pig skin (orange bars), expressed as flux (no close correlation); (b) permeated amount (R2=0.834, n=7); (b) permeability coefficient, logPm and log Kp for Skin PAMPA and pig skin, (R2=0.844, n=9) respectively. All values are mean ± SD.”

13. How was the AUC calculated?

Permeated amount at 6 h, expressed as AUCPAMPA was calculated by integration of the permeability profile between 0 and 6 hours. Text was extended.

“The area under the curve (AUC) was calculated by integration of the permeability profile between 0 and 6 hours using the OriginPro 2019b version.”

14. Conclusion: “…..chemicals dissolved in multiple and widely varying solvent types” please correct as only single chemical was investigated.

Thank you for the remark. Our statement was clarified in the conclusion.

In conclusion, the Skin-PAMPA assay allows the evaluation of the permeability of model compound dissolved in multiple and widely varying solvent types, from highly polar to highly non-polar, as well as mixtures of solvents.”

15. Please include the limitation of this study in the discussion section.

We thank this adequate comment. Text has been extended with the limitations of the assay.

“Great attention had to be devoted to the following factors that are the limitations of this method. Appropriate precise pipetting is essential in this technique. Compounds with too high or too low permeation property cannot be measured. Applying viscous solvents can be challenging because the application of solvents to PAMPA plate is a time-consuming process, so correction with time factor needs to be implemented during the evaluation of the results. Finally, the tension of the solvents can also be a limiting factor since the concentration of high-tension solutions can be modified during the experiment leading to invalid permeability results.”

Reviewer 3 Report

Line 178 – application of piroxicam for membrane integrity study represents non-typical approach when quite sufficient number of recognized standard methods are available for membrane integrity determination. Justification of that would be appreciated.

Line 183 – How the differences in skin thickness were considered in testing procedure.

Line 192 - The reasons for using a receptor fluid for PER to be composed from sodium chloride solution (9 g.L-1) supplemented with 0.25% (v/v) Tween80 have to be provided for clarity.

Line 204 - The specificity of the analytical method was controlled with blank (NaCl, 9 g.L-1) solution evaluation is quite questionable is true specificity (compared to what?) is evaluated. Related question – PER stability during analysis procedure and ability of analytical method to separate PER from potential degradation products.

Line 253 – the statement “Variation i.e. the SD of the permeability values of piroxicam provides a good indication of membrane integrity” has to be clarified for correct understanding.

Line 272 – please clarify why 3 timepoints are considered sufficient.

Line 278 – Clarification of the statement “The different solvents had a significant impact on its permeability” is necessary as earlier the different effect of solvents on PAMPA membrane integrity has been determined.

Line 342 – The meaning of “The classification of solubility using visual assessment” has to be clarified

Line 374 – The penetration process describing parameters are interrelated and they provide information on the same process so the reasoning behind the intention to “identify the best parameter to differentiate between the permeability of the chemical in different solvents” has to be clarified. Alternatively, it could be assumed that those functions determined using PER are not applicable in case of other compounds in the same solvents.

Meanings of abbreviations have to be provided (eg., QC, PG, DMI)

Author Response

1. Line 178 – application of piroxicam for membrane integrity study represents non-typical approach when quite sufficient number of recognized standard methods are available for membrane integrity determination. Justification of that would be appreciated.

Our answer to this question is the same as given for Reviewer #2 at point 8.

The common membrane integrity methods are used to investigate the membrane stability upon storage. In this study our aimed was to reveal the possible damaging effect of organic solvents on the structure of the membrane.  No literature method is available for this purpose therefore, as described in the experimental section, first we visually assessed the membrane condition after the pre-treatment with different solvents. Afterward, the permeability of piroxicam from aqueous solution (a compound used previously in our laboratory for Quality Control of Skin-PAMPA) was measured on pre-treated membrane. We consider that extreme high permeability value would indicate the damage of the membrane. Furthermore, the low standard deviation of the results also shows the good membrane integrity. The text was extended for clarity.

Extreme high standard deviation would indicate the membrane damage. As shown in Fig. 1, the error bars are small (average SD ± 0.08), ethanol (S2) presenting the highest variation (SD: ± 0.23) and thus the largest effect on the membrane, but this SD is still acceptable indicating rather an interaction of ethanol with the membrane than its corruption.

2. Line 183 – How the differences in skin thickness were considered in testing procedure.

Thank you for the remark. This parameter was not studied during the measurements. As reported in the text, the size range was achieved by cutting the dermis below hair follicle

3. Line 192 - The reasons for using a receptor fluid for PER to be composed from sodium chloride solution (9 g.L-1) supplemented with 0.25% (v/v) Tween80 have to be provided for clarity.

The sodium chloride solution in the receptor chamber is an excepted model solution of plasma in the in vitro     penetration test, where the surfactant provides to sink condition.

4. Line 204 - The specificity of the analytical method was controlled with blank (NaCl, 9 g.L-1) solution evaluation is quite questionable is true specificity (compared to what?) is evaluated.

From an analytical perspective, the blank sample was analysed to guarantee the absence of analytical response.

5. Related question – PER stability during analysis procedure and ability of analytical method to separate PER from potential degradation products.

Thank you for this adequate question. Indeed, PER being a biphenol molecule is very sensitive for oxidation, thus all the measurements were performed under light protection. Before the permeability tests, the stability of the compound was controlled by spectrophotometry. It was found that the model compound is stable during 8 hours in all solvents. Although after 24 hours, the PER was degraded in some solvent.

6. Line 253 – the statement “Variation i.e. the SD of the permeability values of piroxicam provides a good indication of membrane integrity” has to be clarified for correct understanding.

See answer at point 1.

7. Line 272 – please clarify why 3 time points are considered sufficient.

These 3 time points on the permeability profiles were feasible for the calculation of flux, since linearity was obtained in this time range in all solvents. See permeability profiles included into the Supplementary material.

8. Line 278 – Clarification of the statement “The different solvents had a significant impact on its permeability” is necessary as earlier the different effect of solvents on PAMPA membrane integrity has been determined.

For membrane integrity study permeability of piroxicam in aqueous solution was measured on solvent pre-treated membrane to demonstrate the membrane integrity (see also our answer to question 1). In case of PER the permeability in different solvents were examined. In the first case the solvent impact on the membrane structure was studied and found no damage (no extreme value obtained in any solvent), while in case of PER the effect of solvent on permeability was significant. The text was clarified.

9. Line 342 – The meaning of “The classification of solubility using visual assessment” has to be clarified

The visual assessment has been revised.

“The classification of solubility based on visual evaluation (of any undissolved particles of the solid) correlated very well with that measured using LC/MS/MS methods for the pig skin assay.”

10. Line 374 – The penetration process describing parameters are interrelated and they provide information on the same process so the reasoning behind the intention to “identify the best parameter to differentiate between the permeability of the chemical in different solvents” has to be clarified. Alternatively, it could be assumed that those functions determined using PER are not applicable in case of other compounds in the same solvents.

We agree with this comment. The correlation analysis between fluxes showed only weak relation, while permeability coefficients correlated very well. In case of PER, we consider the logPm, so the parameter that was normalized for donor concentration differences, to be the best parameter for differentiation between solvents.

11. Meanings of abbreviations have to be provided (eg., QC, PG, DMI)

Thank you for the helpful remark, we inserted the meaning of the abbreviations in the new version of the manuscript.

Round 2

Reviewer 1 Report

I do accept it

Reviewer 2 Report

The manuscript has been improved.